# Beverage Intake and Associated Nutrient Contribution for Aboriginal and Torres Strait Islander Australians: Secondary Analysis of a National Dietary Survey 2012–2013

**DOI:** 10.3390/nu14030507

**Published:** 2022-01-24

**Authors:** Megan A. Rebuli, Gilly A. Hendrie, Danielle L. Baird, Ray Mahoney, Malcolm D. Riley

**Affiliations:** 1Commonwealth Scientific and Industrial Research Organisation (CSIRO) Health and Biosecurity, South Australian Health and Medical Research Institute Level 7, Adelaide 5000, Australia; gilly.hendrie@csiro.au (G.A.H.); danielle.baird@csiro.au (D.L.B.); Malcolm.Riley@csiro.au (M.D.R.); 2Australian e-Health Research Centre, CSIRO Health and Biosecurity, Surgical, Treatment and Rehabilitation Service (STARS) Level 7, Herston 4029, Australia; ray.mahoney@csiro.au

**Keywords:** beverage intake, nutrient intake, sugar-sweetened beverages, nutrition, Indigenous, Aboriginal and Torres Strait Islander, Australian dietary survey

## Abstract

Beverages contribute significantly to dietary intake. Research exploring the impact of beverage types on nutrient intake for Australian Aboriginal and Torres Strait Islander people is limited. A secondary analysis of the Australian Aboriginal and Torres Strait Islander Health Survey 2012–2013 (*n* = 4109) was undertaken. The daily intake, percentage of consumers, and contribution to total nutrient intake was estimated for 12 beverage categories. Beverage intake contributed to 17.4% of total energy, 27.0% of total calcium, 26.3% of total vitamin C, and 46.6% of total sugar intake. The most frequently consumed beverage categories for children (aged 2 to 18 years) were water, fruit juice/drinks, soft drinks, and cordial; and for adults, water, tea, coffee and soft drinks. The primary sources of beverages with added sugar were fruit juice/drinks (for children), tea (for people living remotely), coffee (for adults in metropolitan/regional areas) and soft drinks (for everyone). Actions to modify beverage intake to improve health should maintain the positive nutrient attributes of beverage intake. This analysis of a large-scale national dietary survey provides benchmarking of beverage intake to support program and policy development to modify intake where this is determined as a priority by the community.

## 1. Introduction

Intake of all beverage types in the Australian population has been shown to contribute substantially to total energy intake (16.6% for adults) [1]. Beverages have also been shown to contribute substantially to intake of calcium (28.5%) and vitamin C (22.4%), as well as disproportionately to total sugar intake (35.8%) for Australian adults [1]. As an important component to overall nutrient intake, an in-depth evaluation of the contribution of the beverage category to total energy and nutrient intake can help to target population dietary health interventions and inform relevant policy.

A similar analysis may be relevant for Aboriginal and Torres Strait Islander people (respectfully referred to as Indigenous people in this publication) who made up 3% of the Australian population at the time of the dietary survey [2]. Indigenous people are three times more likely to suffer from diabetes mellitus than non-Indigenous people in Australia (after age-standardisation for those aged 2 years and over) [3], and rates of overweight and obesity are also higher than non-Indigenous people. It is estimated that 42.5% of Indigenous adults are obese and 72.4% overweight or obese [3], compared to 27.5% and 62.8%, respectively, for non-Indigenous people [4].

Behavioural risk factors that contribute to health inequalities between Indigenous and non-Indigenous people include poor nutrition and physical inactivity, which in part may be impacted by social circumstances, environmental stressors, and socioeconomic disadvantage [5]. Descriptive research related to beverage consumption has focused broadly on sugar-sweetened beverages (SSB) for all Australians, and for Indigenous people [6]. This is largely due to the high energy density, high sugar content, and low nutrient value of SSBs, as well as potentially associated negative health outcomes including increased risk of mortality from circulatory disease and digestive disease [7], and increased risk of diabetes, obesity, and dental caries [8,9]. Despite negative health consequences, SSB consumption is common and among a survey of South Australian adults and adolescents in 2014 [10], frequent SSB consumers generally underestimated the sugar content in soft drinks, and were largely unaware of potential negative health outcomes related to high SSB intake [10].

In Australia, dietary intake (including beverage consumption) is influenced by environmental and structural factors that impact on food access and these may differ between population groups. For example, the relative distribution of the Indigenous population and non-Indigenous population varies across the five remoteness categories (major cities; inner regional; outer regional; remote; and very remote) [11]. Close to one-fifth (18.6%) of the Indigenous population live in remote or very remote areas of Australia, compared to only 1.7% of the non-Indigenous population [5,12]. In remote and very remote areas, the range of foods available is reduced and food cost differs significantly from other settings [13].

The 2012–2013 National Aboriginal and Torres Strait Islander Nutrition and Physical Activity Survey (NATSINPAS) [4] conducted by the Australian Bureau of Statistics is the only national comprehensive dietary survey of Indigenous Australians. It was a component of the Australian Aboriginal and Torres Strait Islander Health Survey (AATSIHS) which was, in turn, part of the Australian Health Survey (AHS). A series of national health surveys have been conducted by the Australian Bureau of Statistics (ABS) since 1977; however, only three comprehensive dietary surveys have been a part of these and only one designed specifically for Indigenous Australians. Dietary survey results from the NATSINPAS [4] indicated that Indigenous people living in more remote areas consumed a lower proportion of energy from discretionary foods compared to Indigenous people living in metropolitan and regional areas (35.8% of total energy intake vs. 42.8%) [14] and a lower percentage of Indigenous people in remote areas consumed soft drinks compared to metropolitan/regional areas (31.3% vs. 38.7%) [4]. These would both be considered more beneficial health behaviours reported by those living in remote areas. Possible reasons for the differences in reported soft drink consumption between remote and metropolitan/regional living Indigenous people may be the price of soft drinks and other SSB being significantly higher in remote stores (ranging from 60–400% higher price compared to metropolitan/regional stores) [13]; and the impact of interventions aimed at subsidising and discounting healthier beverages in remote stores such as water and milk while increasing the price of carbonated SSB [15]. Therefore, the geographical differences in SSB intake may be a result of price sensitivity for bought beverages. A scoping review of SSB intake in Australian Indigenous communities [6] called for more detailed examination of the SSB intake data in the NATSINPAS—a broader view of intake of all beverages is expected to provide additional insight.

The aim of this analysis was to describe intake of all beverages and their contribution to energy and nutrient intake of Indigenous people, particularly comparing intake between Indigenous people living in more remote areas to those living in metropolitan/regional areas of Australia, using data from the NATSINPAS [16]. Beverage consumption was examined in adults and children separately because beverage preference and consumption has been shown to differ between these groups [1]. The purpose of the analysis is to provide relevant and specific information to inform policies or programs that aim to modify beverage intake to improve health.

## 2. Materials and Methods

### 2.1. The National Aboriginal and Torres Strait Islander Nutrition and Physical Activity Survey (NATSINPAS)

The NATSINPAS was conducted by the Australian Bureau of Statistics (ABS) in 2012–2013 with 4109 respondents (residing in 2900 households) and a response rate of 79.2%. A detailed description of the sampling framework and data collection methods is available elsewhere [16]. Briefly, the NATSINPAS utilised two 24 h dietary recalls conducted by trained interviewers. The first recall was conducted face to face, and the second recall was attempted (in metropolitan/regional areas only) within two weeks of the first recall via a telephone interview. The Automated Multiple-Pass Method (AMPM) developed by the United States Department of Agriculture (USDA) [17] was adapted for Australian use by the ABS together with Food Standards Australia and New Zealand (FSANZ). This method attempts to maximise the recall of the respondent and was used in conjunction with the NATSINPAS Food Model Booklet [18] and the Bush Tucker Prompt Card [19] to assist respondents in the estimation of portion size and dietary intake.

Remoteness location classifications were developed using the Australian Statistical Geography Standard Remoteness Structure (2016) based on the Accessibility/Remoteness Index of Australia (ARIA+) which measures the remoteness of a point based on the physical road distance to the nearest urban center [11]. The majority (81.4%) of Indigenous people lived in metropolitan and/or regional areas (major cities 37.4%; inner regional 23.7%; outer regional 20.3%) and the remainder (18.6%) lived in the remote areas (remote 6.7%; very remote 11.9%) [12]. This paper refers to the remoteness area categories, major cities; inner regional; and outer regional as metropolitan/regional and the remoteness area categories remote and very remote, as remote.

### 2.2. Secondary Analysis Methodology

Approval to undertake this secondary analysis was given by the CSIRO Social and Interdisciplinary Science Human Research Ethics Committee (CSSHREC 041/20). A condition of the approval was that the Ethics Committee review the manuscript prior to dissemination.

Data from the first dietary recall (day 1) only were used for these analyses. The second day of dietary recall was not used because these were completed by only 18% (*n* = 771) of participants and did not include participants living in remote areas. The use of one day of intake (with weighting factors applied) across a group allowed mean usual beverage intake to be described.

Detailed methodology regarding the categorisation of beverage data from comprehensive food recall data was reported in our previous paper [1]. Briefly, the NATSINPAS dataset provided information regarding all foods and beverages consumed by participants, with one line for each item recalled. Each food or beverage item was identified by an 8-digit hierarchical food code, which allowed beverages (fluid foods that are drunk) to be distinguished by the first 3 digits of the code. The dataset also provided a food combination code, identifying where separate food items were combined and consumed as a single food (i.e., a sandwich or cereal and milk). The first step of classifying beverage consumption was to identify and exclude fluids which were combined and eaten as a food (e.g., milk in cereal) rather than consumed as a beverage. The combination code was also used to correctly allocate beverages to a category where more than one fluid was combined prior to consumption (e.g., milk added to tea was categorised as tea rather than plain milk), and to capture non-fluid additions to beverages (i.e., sugar in tea) (Table 1).

Most fluid foods that were not identified with a combination code were assumed to be consumed as a beverage. Soups were not categorised as a beverage. Nutrient contribution was estimated from the individual beverage and aggregated across the surveyed day for each participant into the relevant beverage category, and into beverages overall. The specific beverage category classifications are listed in Table 1.

### 2.3. Statistical Analysis

This secondary analysis focused specifically on beverage intake in the Indigenous population, examining patterns of intake by age group, aggregated age groups (children 2–18 years and adults 19 years and above), and area of residence (metropolitan/regional and remote). For each beverage category, estimates were calculated for the percent of the population group who consumed each beverage category and the mean daily intake (10th and 90th percentile) in grams. The mean contribution of each beverage category to total nutrient intake was estimated for energy, calcium, vitamin C, and total sugar (excluding supplements). If 20 subjects or less consumed the beverage category within any demographic group, estimates were considered unreliable and were not reported.

Prior to analyses, population weights based on age, gender, and residential area (provided in the NATSINPAS dataset) were applied. The estimates were further weighted for the day of the week the survey occurred because the percentage of subjects reporting their intake for a Friday or Saturday (i.e., 24 h dietary recalls conducted on Saturday and Sunday) were under-represented. The population weights were scaled to the size of the sample for inferential statistics.

Descriptive statistics were used to report the number and proportion of consumers of beverage categories, the mean beverage intake weight, and the mean percentage contribution of beverage categories to nutrient intake, by age and remoteness. The difference between remote and metropolitan/regional area for weight of beverage intake and contribution to total nutrient intake was tested for statistical significance using the Mann–Whitney U test with no adjustment for multiple comparisons. The difference between the proportion of consumers in remote and metropolitan/regional areas within beverage categories was assessed using a chi-square test.

A *p*-value of less than 0.05 was taken to indicate statistical significance. Statistical analyses were performed using the IBM SPSS statistical software package version 25 (SPSS Inc., Chicago, IL, USA).

## 3. Results

There were 4109 participants in this survey, and 56.4% were considered to live in the over-sampled remote areas. The proportion of the sample that were female was similar in the remote and metropolitan/regional subgroups (56.1% and 55.5%, respectively). The mean age of participants was 30 years, with 36.4% aged between 2 and 18 years and 63.6% aged 19+ years (Table 2). The proportions represented in this surveyed group differs from the proportions in the Indigenous population in Australia primarily because of intentional over-sampling of participants from remote regions.

### 3.1. Population Beverage Consumption

Almost all (99.5%) survey participants reported consuming at least one beverage on the day of the survey (Appendix A). The median daily number of beverage categories consumed per person was two for people living in remote areas and three for people living in metropolitan/regional areas, and the 90th percentile was four beverage categories in remote and metropolitan/regional areas. Children (2–18-years old) consumed a median of two beverage categories, and adults (19 years and above) consumed three. The total mean intake of beverages was 2061 g, with children reporting 1501 g and adults 2446 g (Appendix A). There was no difference in overall mean beverage intake between those living in metropolitan/regional and remote areas (2042 g vs. 2137 g, *p* = 0.285; Appendix A).

### 3.2. Beverage Consumption by Type

The most commonly consumed beverage category was water, with 83.8% of participants consuming water on the day of the survey (Appendix A). The proportion of participants consuming water ranged from 74.5% of 2–3-year-olds to 87.9% of 4–8- and 71+-year-olds living in metropolitan/regional areas, and from 83.4% of 51–70-year-olds to 94% of 4–8-year-olds living in remote areas. The mean water intake was significantly greater in remote vs. metropolitan/regional (1251 g vs. 1177 g, *p* = 0.015), more specifically for children (1059 g vs. 935 g, *p* = 0.031) but not adults (1379 g vs. 1366 g, *p* = 0.441). The only age group for which water consumption was greater in metropolitan/regional areas was 2–3-year-olds, with remote children in this age group reporting 544 g compared to 749 g for metropolitan/regional children (*p* = 0.036).

After water, fruit juice/drinks (37.4%) and soft drinks (30.9%) were consumed by the greatest percentage of children, followed by cordial (20.9%) and plain milk (19.5%) (Figure 1). The mean consumption (among consumers) was highest for fruit juice/drinks (560 g), followed by soft drinks (463 g), cordial (313 g), and milk (292 g) (Appendix A). A higher percentage of children in remote areas compared to metropolitan/regional areas consumed tea (23.1% vs. 5.1%, *p* < 0.001), whereas a higher percentage of children in metropolitan/regional areas consumed soft drinks and milk (soft drink: 34.1% vs. 28.6%, *p* = 0.002; milk: 22.3% vs. 17.4%, *p* = 0.001).

More than half the adults surveyed (54.7%) consumed tea, and over one-third consumed coffee (36.1%) and soft drinks (33.8%) (Figure 2, Appendix A). The consumption of tea was more common in adults living in remote compared to metropolitan/regional areas (66.7% vs. 39.3%), whereas coffee consumption was more common in metropolitan/regional areas (metropolitan/regional: 49.9% vs. remote: 25.2% (Figure 2, Appendix A). Consumption of alcoholic beverages was more common in men than women (26.3% vs. 11.8%, *p* < 0.001) and the mean intake by male consumers was double that of female consumers (2104 g vs. 996 g, *p* < 0.001). Consumption of alcoholic beverages by adults was more frequent in metropolitan/regional areas (21.4% vs. 14.9%); however, adults who consumed alcoholic beverages and lived in remote areas consumed a greater amount than those living in metropolitan/regional areas (2554 g vs. 1593 g, *p* = 0.001) (Appendix A).

### 3.3. Contribution of Beverages to Energy and Nutrients

On average, overall beverage intake contributed 17.4% of total energy intake (Appendix A). There were differences in the contribution of beverages to total energy and nutrient intakes between adults and children; however, there were fewer and smaller differences between participants living in remote and metropolitan/regional areas (Figure 3). The percentage of total energy from beverages was 19.5% for adults and 14.3% for children, and higher, but not statistically significantly higher, in metropolitan/regional than remote areas for all ages (17.6% vs. 16.4%, *p* = 0.07). Beverages contributed as much as 21.6% of total energy for 31–50-year-olds, with the contribution being 22.1% for those living in metropolitan/regional areas vs. 19.9% for those in remote areas (*p* = 0.023, Appendix A).

Beverages contributed 27.0% to total calcium intake, 26.3% to total vitamin C intake and 46.6% to total sugar intake overall (Appendix A). The contribution of beverages to total calcium intake was 29.4% for adults and 23.6% for children (Figure 3, Appendix A). Between age groups, 31–50-year-olds had the greatest percentage contribution of beverages to calcium among adults (32.7%), and only 2–3-year-olds had a higher percentage of calcium coming from beverages (35.7%, Appendix A). Despite only small differences in the contribution of beverages to total calcium intake between remote and metropolitan/regional areas overall, the difference was significant for 2–3-year-olds only (*p* = 0.042, Appendix A).

Beverages contributed 30.7% of children’s and 23.2% of adult’s total vitamin C intake. Among children, this contribution was highest in 4–8-year-olds (34.3%). For adults, it was highest in 19–30 years (30.0%) and declined with age, with beverage intake in 71+ years contributing only 6.9% of total daily vitamin C intake. There were no significant differences in the contribution of total vitamin C intake from beverages between those living in remote versus metropolitan/regional areas.

The contribution of beverages to total sugar intake ranged from 28.0% in 71+-year-olds to 56.0% in 19–30-year-olds. The contribution of beverages to sugar intake was similar for remote and metropolitan/regional living children. The contribution of beverages to total sugar intake was highest for 14–18-year-olds (51.5%) followed by 2–3-year-olds (42.0%, Appendix A). Overall remote living adults, and particularly in the 51–70-year-old group, consumed a significantly higher proportion of sugar from beverages than metropolitan/regional living adults of the same age (19+ years: 51.3% vs. 49.1%, *p* = 0.041; 51–70 years: 39.5% vs. 34.5%, *p* = 0.015).

### 3.4. Contribution of Beverage Categories to Energy and Nutrients

The contribution of beverage categories to energy and nutrients and the proportion of participants consuming beverages are shown in Figure 1 (children) and Figure 2 (adults) by metropolitan/regional and remote areas of residence.

The beverage categories that contributed most to children’s energy intake, and to sugar intake, were soft drinks (3.2% energy, 12.5% sugar) and fruit juice/drinks (2.9% energy, 10.9% sugar) (Figure 1a,d, Appendix A). For all children, the contribution of fruit juice/drink to energy intake was higher in metropolitan/regional than remote children (fruit juice/drink: 3.0% vs. 2.3%, *p* = 0.039) (Figure 1a, Appendix A).

Plain and flavoured milks contributed significantly more to the calcium intake of metropolitan/regional children than remote living children, (plain milk: 7.6% vs. 6.0%, *p* = 0.002; flavoured milk: 7% vs. 4.3%, *p* = 0.015) (Figure 1b, Appendix A). Tea was the second-highest contributor to calcium intake for remote living children (4.7%), after plain milk; however, tea was consumed by fewer metropolitan/regional children (5.1% vs. 23.1%) and contributed a negligible amount to their calcium intake (0.5%, *p* < 0.001 vs. remote children’s calcium intake).

Fruit juice/drink was the primary beverage type contributing to children’s vitamin C and accounted for approximately one-fifth of overall intake (remote: 21.0% vs. metropolitan/regional: 24.8%, *p* = 0.024) (Figure 1c; Appendix A).

All beverages contributed to just under half of children’s total sugar intake (metropolitan/regional: 42.2%, remote: 42.1%, *p* = 0.93) (Appendix A). Soft drinks, fruit juice/drinks, cordial, and tea (for remote living children only), each contributed to at least 5% of overall sugar intake (Figure 1d, Appendix A). The top three contributed similar proportions of sugar to intake for remote and metropolitan/regional dwelling children, with the exception of soft drink and cordial in 9–13-year-old children, where soft drinks contributed significantly more to the intake of metropolitan/regional children (15.4% vs. 9.5%, *p* = 0.023) and the reverse for cordial in this age group (3.9% vs. 8.6%, *p* = 0.045). Contribution of beverages to sugar intake was also significantly higher in metropolitan/regional than remote 14–18-year-olds (11.9% vs. 7.1%, *p* = 0.034) (Appendix A). Tea contributed to 6.3% of remote children’s sugar intake, compared to 0.7% of metropolitan/regional children (*p* < 0.001).

The beverage categories that contributed most to energy intake for adults were alcoholic beverages (5.3%) and soft drinks (3.9%). For adults living remotely, tea was the second-highest contributor to energy intake (4.6%), significantly higher than the contribution from tea for metropolitan/regional living adults at 1.5% (*p* < 0.001) (Figure 2a, Appendix A). There were differences between remote and metropolitan/regional living adults regarding consumption of tea and coffee. Tea was consumed by 66.7% of adults in remote areas and contributed 4.6% of energy, compared to 39.3% of adults and 1.5% of energy for metropolitan/regional areas. Conversely for coffee, 49.9% of adults in metropolitan/regional areas consumed coffee and its contribution to total energy was 3.7%, compared to 25.2% of adults and 1.4% of energy in remote areas. As a consequence, contribution of tea to total calcium intake in remote areas was higher than metropolitan/regional areas (11.1% vs. 3.5%, *p* < 0.001), and the contribution of coffee to calcium was higher in metropolitan/regional areas than remote areas (9.9% vs. 4.2%, *p* < 0.001) (Figure 2b, Appendix A).

Beverages that contributed most to total vitamin C intake were fruit juice/drinks (metropolitan/regional: 13.5% vs. remote: 9.6% *p* = 0.01) followed by alcohol-containing beverages (metropolitan/regional: 5.9% vs. remote: 6.4%, *p* = 0.147) for adults (Figure 2c).

For adults living in remote areas, tea (66.7% consumed; contributing 16.3% of total sugar), soft drinks (30.1% consumed; contributing 14.9%) and cordial (13.6% consumed; contributing 4.8%) made the greatest contributions to total sugar intake. This contrasted with adults living in metropolitan/regional areas, where soft drinks (38.6% consumed; contributing 14.7% of total sugar), coffee (49.9% consumed; contributing 9.9%) and fruit juice/drinks (15.5% consumed; contributing 6.0%) contributed most to total sugar intake (Figure 2d). The contribution of tea, coffee, and fruit juice/drink to sugar intake varied significantly dependent on metropolitan/regional and remote areas (*p* < 0.001 for tea and coffee, *p* = 0.009 for fruit juice/drink) (Appendix A).

## 4. Discussion

This paper described the consumption of beverages and their contribution to nutrient intakes in the most recent (and only) nationwide comprehensive dietary survey of Indigenous Australians [15]. The relevance of beverages to overall dietary intake was apparent, with almost all participants reporting consumption of at least one type of beverage on the day of the survey. Beverages provided a substantial proportion of overall energy and nutrient intake—energy (17.4%), sugar (46.6%), calcium (27.0%), and vitamin C (26.3%) on the day of the survey. Adults consumed a greater volume of beverages than children, and beverages accounted for a greater contribution to energy intake for adults. Adults also chose different types of beverages, for example, adults consumed tea and coffee whereas children preferred fruit juice and soft drinks. The fact that beverage preferences differ across life stages supports the importance of establishing healthy beverage choices early in life.

There were also differences in beverage choices and their contribution to energy and nutrients between metropolitan/regional and remote living survey participants, suggesting area of residence is an important influence on beverage consumption patterns. Differences in consumption might be attributed to a range of factors, such as environment, infrastructure, preference, availability, pricing, knowledge and perception of healthy beverages, and social and cultural influences. Nonetheless, these data highlight the importance of targeted health promotion and education efforts based on individual factors such as age as well as contextual factors such as location of residence.

After water, the top four most commonly consumed beverage categories reported for Indigenous children were fruit juice/drinks, soft drinks, cordial, and plain milk, and for adults were tea, coffee, soft drinks, and alcoholic beverages. Similar choices were seen in the 2011–12 Australian National Nutrition and Physical Activity Survey (NNPAS) of all Australians, with the exception of children in the all Australian survey, who consumed flavoured milk more commonly than cordial [1]. The contribution of beverages to total nutrient intake was similar between the surveys for energy (17.4% for the Indigenous survey vs. 15.8% for the all Australian survey), calcium (27.0% vs. 28.0%), and vitamin C (26.3% vs. 23.7%); however, the overall contribution of beverages to sugar intake was higher (46.6% vs. 35.7%) [1]. While mean total sugar intake was marginally higher for Indigenous people (6% for all age groups) compared to the result for all Australians, mean total dietary calcium intake was 17% lower for Indigenous people compared to the result for all Australians [4]. Intake of total sugars for Indigenous people was 22.7% of energy for children 2 to 18 years, and 20.5% of energy for adults 19 years and over [4]. This high level of intake is similar to non-Indigenous Australians and is the motivation for the Australian Dietary Guideline to ‘Limit intake of foods and drinks containing added sugars such as confectionary, sugar-sweetened soft drinks and cordials, fruit drinks, vitamin waters, energy and sports drinks’ [20].

Indigenous children attained a higher proportion of their vitamin C from beverages than adults, largely attributable to fruit juice/drinks, whereas adults attained more of their daily calcium from beverages than children, with the highest beverage sources being coffee for metropolitan/regional living adults and tea for remote living adults.

Tea intake is strikingly higher for Indigenous people living in remote regions compared to Indigenous people living in metropolitan/regional areas. The high consumption of tea in remote living Indigenous people seems to be unique to this population group as it was not observed in metropolitan/regional living Indigenous adults or the survey of all Australians [1]. To our knowledge, the survey of all Australians has not been analysed by measures of remoteness, so it is not known whether this beverage pattern also applies to non-Indigenous Australians. The purpose of this manuscript is to describe beverage intake amongst Indigenous people and its impact on intake of selected nutrients, rather than attempt to explain why beverage intake is as it is. The latter task is almost certainly important to efforts to change beverage intake, and it is noted that while culture is central to explaining differences in beverage intake [21], it has also been suggested that the preference for sweetened tea amongst remote living Indigenous people is an example of an introduced dietary habit (when sugar and tea were provided as rations) conserved over many decades [22]. It is also noted that tea does not have challenging storage requirements and is relatively inexpensive to transport which may also contribute to its popularity. It is stressed that studies involving local communities should be undertaken to understand food intake patterns.

Consuming foods and beverages with high quantities of added sugar is known to increase the likelihood of developing chronic disease [10]. Primary sources of beverages with added sugar in this survey were fruit juice/drinks (for children), tea (for people living remotely), coffee (for adults in metropolitan/regional areas) and soft drinks (for everyone). For people living remotely, tea contributed almost as much to total sugar intake as soft drinks (12.5% compared to 13.4%). Coffee also contributed to the total sugar intake of metropolitan/regional dwelling adults, however to a lesser extent than tea or soft drinks. The National Aboriginal and Torres Strait Islander Health Survey (NATSIHS) 2018–2019 [23], reported that sugar-sweetened and diet drinks were consumed by approximately 7 in 10 Indigenous people, and the proportion of people consuming these beverages was higher in remote areas than non-remote areas (77% vs. 69%). Noting that there are differences between the dietary survey and beverage categorisation methodology between the 2018–2019 survey and the survey data utilised in the current secondary analysis, these results consistently suggest that location may impact beverage intake. Many calls for public health interventions targeting a reduction in sugar intake in Australia (generally and specifically to Indigenous communities) have focused on sugar-sweetened beverages such as soft drink and fruit juice. The data presented here indicate that adding sugar to hot beverages is a significant source of sugar in some Indigenous population groups.

Calcium is important for bone density, and adequate dietary calcium in childhood, teenage, and early adult years is essential for prevention of bone loss-related disease in later life [24]. Milk consumed as a beverage has been shown to be an important contributor to total calcium intake for Australian children [1]. The proportion of children reported to consume milk was lower in Indigenous children than in the Australian population survey; and lower still for Indigenous children living in remote areas, with twice as many children living in metropolitan/regional areas consuming milk. Indigenous children in remote areas consumed approximately half a cup less on average than their Indigenous metropolitan/regional and Australia wide counterparts. As a result, plain milk contributed to 6.0% of total calcium intake for Indigenous children living in remote areas, compared to 7.6% of total calcium intake for Indigenous children living in metropolitan/regional areas, and 9.9% for children Australia wide [1]. This occurs in a context where total mean dietary calcium intake was substantially poorer (i.e., 17% less) for Indigenous people than non-Indigenous people [4], and all age groups above 9–13 years (male and female) have an average calcium intake that is less than the estimated average requirement for calcium [4]. While children are more likely to drink plain milk, adults are more likely to drink milk with tea and coffee. Consistent with the pattern for Indigenous children, Indigenous adults living remotely consumed 80–100 mL less plain milk on average than their Indigenous metropolitan/regional counterparts.

While beverages make a strong contribution to vitamin C intake, this nutrient is not in short supply among Australian Indigenous people—in all male and female age groups, the average vitamin C intake was well above the estimated average requirement [4].

Alcoholic beverages have health guidelines separate to the dietary guidelines, and are often considered separately to other elements of dietary and beverage intake. The rationale for their inclusion as a beverage category in this analysis is that they contribute to nutrient intake (particularly energy intake), other beverages are mixed with them on occasion, and pragmatically beverages might be substituted for each other. There may be compensation for changing intake of one category of beverages across demographic groups which might not be apparent if only a subset of beverages is analysed. The Australian National Health and Medical Research Council (NHMRC) risk reduction guidelines [25] for alcoholic beverage consumption for adult Australians states that the less you drink, the lower your risk of harm from alcohol. Where lifetime risk is benchmarked as consuming more than two standard drinks per day on average and single occasion risk as consuming more than four standard drinks on any one occasion. The proportion of all Australian adults who exceed the lifetime risk guideline decreased from 19.1% in 2013 to 18.0% in 2016, while 39% of all Australian adults exceeded the single occasion risk guideline [26]. The NATSIHS 2018–2019 reported that the percentage of Indigenous adults (18+ years) consuming alcoholic beverages at the single occasion risk level was reduced from 57% in 2012–2013 to 54% [23]. The number of people consuming alcoholic beverages at the lifetime risk level had not changed since 2012–2013, remaining at 20% in each survey. The data presented in this current study focused on differences in consumption patterns by location of residence. These data suggested that a greater proportion of Indigenous adults living in metropolitan/regional areas consumed alcoholic beverages compared to those living in remote areas (21.4% vs. 14.9%); however, when alcoholic beverages were consumed by Indigenous people living in remote areas, they were typically consumed in a greater volume. The difference in the amount consumed on the day of the survey was in the order of one litre more in remote compared to metropolitan/regional adults (2554 g vs. 1593 g). Comparing these results to the Australia wide dietary survey results, suggests that alcoholic beverages are consumed by a much smaller proportion of Indigenous adults living in remote areas than Australian adults generally (14.9% vs. 33%) [1]. However, Indigenous adults living in remote areas who consumed alcoholic beverages, consumed almost three times as much as Australian adults generally who consumed alcoholic beverages (2554 g vs. 806 g). Further to this, the NATSIHS 2018–2019 reported that 26% of Indigenous adults had not consumed an alcoholic beverage in the last 12 months (or had never consumed alcoholic beverages), an increase from the 2012–2013 survey (23%). This is a higher percentage of “teetotallers” compared to the 22.2% of all Australian adults reported to have not consumed alcoholic beverages in the past year for 2016 [26]. Alcoholic beverage intake is not recommended for health reasons; however, they make a significant contribution to energy intake for Indigenous adults, and some contribution to calcium, vitamin C and sugar intake. While some Indigenous adults consume alcoholic beverages at a higher risk level, a large percentage do not consume alcoholic drinks at all.

Improving the health of Indigenous people has been on the national health agenda for decades [27], and it is generally agreed that culturally sensitive interventions are needed that are co-designed with and led by the Indigenous community. A good example of this is in Cape York Australia, where the Apunipima Cape York Health Council, together with the local hospital and health service and Cairns Public Health unit, developed a Food and Nutrition Strategy to prioritize nutrition services and initiatives in the region for Indigenous people in remote areas [28]. A key aspect of this strategy is the necessity for Indigenous community-controlled ‘grass roots’ approaches to delivery. One intervention launched in this remote region of Australia is the “Sugary Drinks Proper No Good—Drink More Water Youfla”. As its title implies, this intervention targets the reduction in sugary drink consumption and increased consumption of water, promoting it as ‘the healthiest drink since the beginning of time’. This campaign was developed with Indigenous people, using a sporting legend ambassador, local children, and respected elders to help deliver this important message. While results of this program are not available yet, results from the current analysis suggests that sugary beverages are only part of the overall impacts of beverages on diet quality of this population. Public health promotion activities in recent years that target beverage consumption tend to focus on reducing the intake of pre-packaged SSB; however, as identified in this paper, additional messages could include increasing the consumption of calcium-containing beverages, reducing alcoholic beverage intake, and reducing the addition of sugar to drinks like tea and coffee. Our findings show that there is a need for a shift in thinking to consider culturally appropriate interventions based on ‘where you are’ as opposed to ‘who you are’, and a deeper exploration into motivation for current beverage intake practices.

A review by Gwynn et al. [29], looking at nutrition interventions targeting the improvement of diet and health in Indigenous people, found interventions that were store-based and included community health promotion were more successful in very remote locations. Another review [6], looking specifically at interventions targeting the reduction in sugar-sweetened beverages, found the majority of interventions were delivered in remote areas, were community-driven initiatives, and implemented through community-owned stores. This review found the effect of price discounts on healthy beverages in the order of 10–20% did not reduce SSB sales; however, the greatest impact on SSB sales was an intervention (directed by the community) that removed the three highest-selling SSB from stores in their community, indicating in-store availability may be a greater driver of SSB sales than price. There is a lack of interventions in this space targeting non-remote Indigenous people [6], and this work further highlights the need to better understand the different demographic subgroups of the Indigenous population in helping to develop more culturally sensitive and effective interventions in the future.

The NATSINPAS 2012–2013 is the largest comprehensive dietary survey for Indigenous people to date, it was conducted throughout Australia, and included remote communities and groups of people who are rarely surveyed. Additionally, the overarching Australian Aboriginal and Torres Strait Islander Health Survey 2012–2013 (AATSIHS) was developed and implemented through collaboration with advisory groups encompassing Indigenous members [30]. The ABS employed trained interviewers to conduct face-to-face interviews, who cooperated with Indigenous communities and families to minimise misreporting.

An additional strength to this study is the comprehensive approach to categorising all beverages. Liquid foods consumed with other foods (such as milk with cereal) were excluded as not being beverages, and considerable care was taken to describe the composition of beverages in their consumed form (for example water with added cordial concentrate was classified as cordial, and additions to beverages were included such as table sugar added to a beverage). This differentiates it from other analyses using data from this nutrition survey. Another strength of this study is the additional weighting applied to the ABS population weighting to correctly account for the proportion of days with recorded dietary intake. Beverage type and quantity are likely to vary between the days of the week and weekend days; thus, the analysis of the population beverage intake is likely to be biased where Friday and Saturday dietary intake is under-represented.

Limitations of this study include inherent error in 24 h dietary recall data, such as under reporting total food intake, and the difficulty of quantitatively reporting food intake where the food is not ‘portion packaged’ (such as water from a tap). A single day of intake only was used for each study subject; therefore, while it is possible to estimate average usual group intake, the percentage of a group who reported consuming from a beverage category over a single day is lower than the equivalent percentage over a longer period. The beverage categories were constructed to aggregate beverages that are used in a similar way while maintaining a relatively small number of categories. Therefore, sugar-sweetened soft drinks are included with artificially sweetened soft drinks, fruit juices are all included together regardless of their dilution, and alcoholic drinks are aggregated regardless of their alcohol concentration. The aim of the analysis was to describe the intake of all beverages and their contribution to nutrient intake—since the range of beverages is large, categorisation is necessary to assist in comprehension. The survey was conducted 9 years ago in 2012–2013; however, the survey is unique in its size and national coverage and provides benchmark information for future dietary surveys at a local or national level.

## 5. Conclusions

Beverages make an important contribution to dietary intake of Indigenous people at a similar level to non-Indigenous Australians. Many beverages, for example sugar-sweetened beverages or alcoholic drinks, are not recommended for consumption by health authorities. This analysis highlights that concern about sugar in beverages should take account of discretionary sugar added to beverages such as tea and coffee in addition to pre-mixed drinks such as soft drinks. Interventions to modify beverage intake should recognize that some positive nutrients are contributed by beverages. There are some differences in beverage intake between Indigenous people living in remote areas and those living in metropolitan/regional areas. The reasons for this are unclear and may be important for efforts to encourage beverage intake consistent with better health. Many inter-related factors influence food and beverage preference and consumption, and prerequisites for successful programs to modify intake are likely to be community prioritization of a need to change, and co-design of programs led by Indigenous people. This analysis of a large-scale national dietary survey provides benchmarking of beverage intake to support program and policy development.

## Figures and Tables

**Figure 1 nutrients-14-00507-f001:**
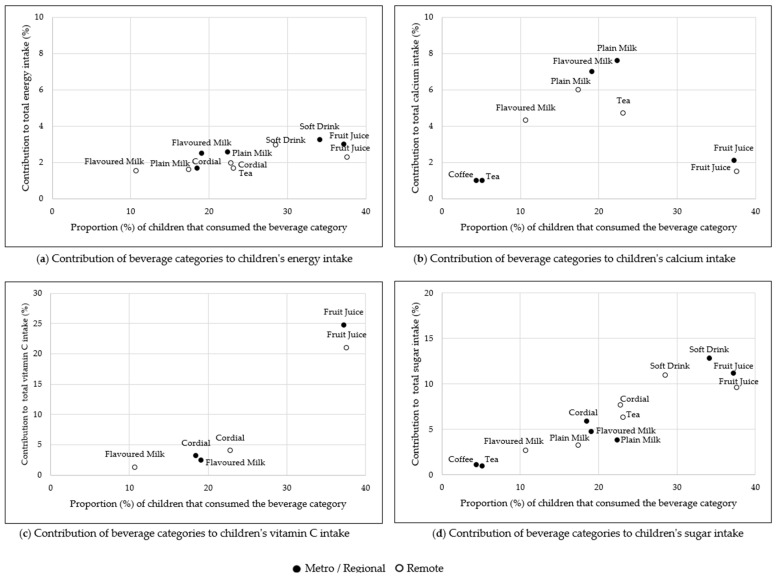
The contribution of beverage categories to energy (kJ) and nutrient intake for Indigenous Australian children (2 to 18 years) living in metropolitan/regional and remote areas by the prevalence of consumption on the day of the survey. (**a**) shows the contribution of beverage categories to energy intake, (**b**) shows the contribution to calcium intake, (**c**) the contribution to vitamin C intake and (**d**) shows the contribution to sugar intake. Beverage categories contributing less than 1% to the relevant intake are not shown.

**Figure 2 nutrients-14-00507-f002:**
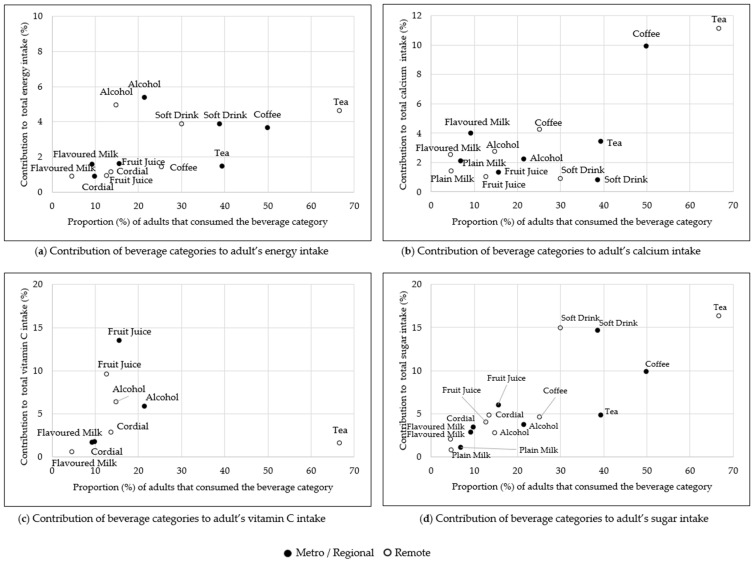
The contribution of beverage categories to energy (kJ) and nutrient intake for Indigenous Australian adults (19+ years) living in metropolitan/regional and remote areas by the prevalence of consumption on the day of the survey. (**a**) shows the contribution of beverage categories to energy intake, (**b**) shows the contribution to calcium intake, (**c**) shows the contribution to vitamin C intake and (**d**) shows the contribution to sugar intake. Beverage categories contributing less than 1% to the relevant intake are not shown.

**Figure 3 nutrients-14-00507-f003:**
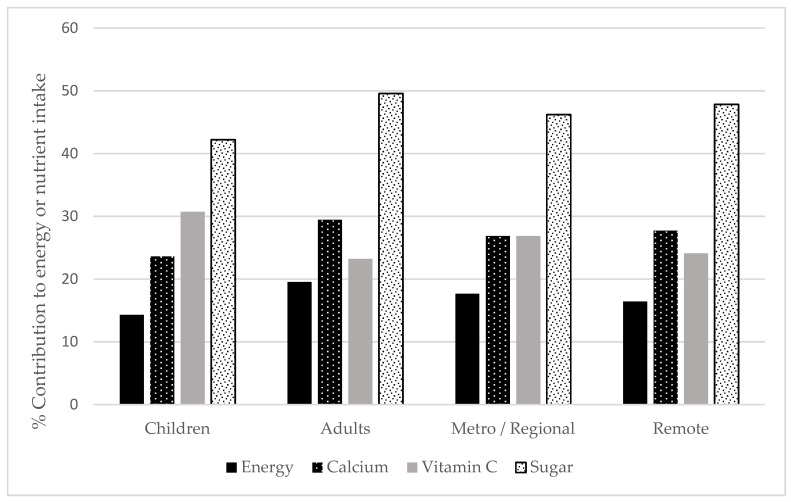
The percentage contribution of total beverage intake to energy (kJ) intake, calcium intake, vitamin C intake and sugar intake for Indigenous Australian children and adults, living in metropolitan/regional and remote areas.

**Table 1 nutrients-14-00507-t001:** Description of beverage category membership.

Beverage Category	Description
Alcoholic beverages	All beverages including any alcohol content. Mixers are included in the category, but any alcoholic beverage used as an ingredient in food is not included.
Tea	All home brewed tea plus all additions (milk, sugar, water) are included.
Coffee	Hot coffee plus all additions (milk, sugar, water) are included. Cold coffee flavoured milk beverages are included in flavoured milks.
Soft drink	All flavoured carbonated beverages whether sugar sweetened or sweetened with other sweetening agents.
Cordial	All flavoured drinks made up with water from a concentrate.
Energy drinks	Energy drinks and electrolyte (‘sport’) drinks.
Fruit juices and drinks	All fruit and vegetable juices and drinks (non-carbonated), regardless of their dilution.
Plain milk	Plain white milk without flavouring or additives, regardless of fat content. Milk used as an ingredient for food is not included; milk as an ingredient of beverages was included in the respective categories.
Flavoured milk	All flavoured milk (hot or cold) whether as purchased or produced through adding powdered flavouring to milk.
Milk alternatives	Plain or flavoured dairy milk alternatives such as soy milk and nut milks, not used in food or as an addition to another beverage category.
Other beverages	Powdered flavourings with water, probiotic drinks, breakfast cereal beverages, protein and supplement powders.
Water	All water consumed as a drink, but not included in any other beverage category. Includes carbonated and still water.

**Table 2 nutrients-14-00507-t002:** Demographic characteristics of participants in the National Aboriginal and Torres Strait Islander Nutrition and Physical Activity Survey 2012–2013.

	Survey Sample		Metropolitan/Regional Living		Remote Living	
	*n*	%	*N*	%	*n*	%
Total	4109	100	1792	43.6	2317	56.4
Sex (*n*)						
Male	1814	44.1	797	44.5	1017	43.9
Female	2295	55.9	995	55.5	1300	56.1
Mean age (years)	30		30.1		30.1	
Age category						
2–3	240	5.8	102	5.7	138	6.0
4–8	515	12.5	214	11.9	301	13.0
9–13	409	10.0	176	9.8	233	10.1
14–18	332	8.1	153	8.5	179	7.7
All children (2–18)	1496	36.4	645	36.0	851	36.7
19–30	722	17.6	332	18.5	390	16.8
31–50	1098	26.7	482	26.9	616	26.6
51–70	722	17.6	300	16.7	422	18.2
71+	71	1.7	33	1.8	38	1.6
All adults (19+)	2613	63.6	1147	64.0	1466	63.3

## Data Availability

The data presented in this study are available on request from the Australian Bureau of Statistics.

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
