# Peer review of "Beverage Intake and Associated Nutrient Contribution for Aboriginal and Torres Strait Islander Australians: Secondary Analysis of a National Dietary Survey 2012–2013"

_nutrients, 2022, doi:10.3390/nu14030507_

Round 1

Reviewer 1 Report

Dear authors,

The aim of this analysis was to describe intake of all beverages and their contribution to energy and nutrient intake of Indigenous people, particularly comparing intake between Indigenous people living in more remote areas to those living in metropolitan/regional areas of Australia, using data from the National Aboriginal and Torres Strait Islander Nutrition and Physical Activity Survey (NATSINPAS).

Although the purpose of this analysis was to provide relevant and specific information to policies or programs that aim to modify beverage intake to improve health, additional information should be added to clarify and enhance the relevance of the manuscript:

- Authors should contextualize the dietary habits and food cultural insights from Indigenous people and their major difference regarding the Australian population in general.

- Authors should contextualize the NATSINPAS into Australian health surveys plan.

- Authors should explain the main reasons to analyze data from a national dietary survey from 2012-13 and explain its relevance in now-a-days.

- Authors should map the relative distribution of the Indigenous population across the five remoteness categories: major cities; inner regional; outer regional; remote; and very remote.

- Results should also be analysed considering the recommendations by the Australian Dietary Guidelines (i.e., compared the mean dietary intake of the nutrient to Australian Dietary Guidelines estimate average requirement, for each life-stage group).

- Did authors expect the fact that 31-50-year-olds had the greatest percentage contribution of beverages to calcium among adults (32.7%), and only 2-3-year-olds had a higher percentage of calcium coming from beverages (p. 8)?

- Authors should explain why the high consumption of tea in remote living Indigenous people seems to be unique to this population group as it was not observed in metropolitan/regional living Indigenous adults or the survey of all Australians (p. 11). Is there any relation with proximity of tea plantation area?

- Authors should explain why Indigenous people living in remote areas that consumed alcoholic beverages, consumed almost three times as much as Australian adults generally who consumed alcoholic beverages.

Author Response

We thank the reviewers for their comments on our manuscript, and acknowledge the time and thought that has gone into their reviews. We have made a number of amendments to the manuscript that has resulted in its improvement, and we appreciate the contribution from the reviewers.

The authors have once again reviewed the manuscript for English language expression, and have re-numbered the references to include new citations.  

With respect to specific comments:

Reviewer 1:

  1. Indigenous people in Australia are a diverse group who make up a small percentage (3%) of the Australian population. They have been subject to colonisation for more than 200 years with the majority (more than 80%) now living in cities and major urban centres. Dietary habits have been studied for small non-representative groups, and the dietary survey that is the subject of this manuscript is the first comprehensive national dietary survey of Indigenous Australians. We are unable to appropriately contextualise dietary habits or differences to the general Australian population except as shown by this survey and referred to throughout.
  2. Contextual information about the NATSINPAS has been added to lines 66-71.
  3. The 2012-13 National Aboriginal and Torres Strait Islander Nutrition and Physical Activity Survey is the most recent national dietary survey of Aboriginal and Torres Strait Islander people (and, in fact, the only such survey to date). Aboriginal and Torres Strait Islander people are a small percentage of the Australian population (estimated to be 3.0% of the Australian population on June 30 2011 – the closest census date*) and therefore the sample size in national Australian surveys does not allow detailed analyses of dietary intake. In any case, the most recent Australian National survey was in 2011-12. While general results from the dietary survey have been published by the Australian Bureau of Statistics (and referred to in the introduction), this analysis is specifically focussed on beverages using methods developed by our team. The NATSINPAS is the only survey available to examine dietary data from Indigenous people on a national basis providing benchmark information for potential future surveys, and allowing comparison by regional living situation (remote compared to metropolitan/regional).  We have added information on the percentage of the Australian population who are Indigenous Australians (lines 35-36), emphasised the unique nature of the NATSINSAP (lines 64-66) and included the age of the dietary data as a weakness (lines 543-545).   
  4. The information about the relative population percentages in each geographic region is provided in lines 107-110. The oversampling in the remote areas (lines 175-177) is adjusted for in the final estimates by the use of population weights (lines 153-158).
  5. We appreciate the point made by the reviewer – that contribution to estimated intake does not address the potential impact of the nutrients provided from beverages since they may be in excess to requirements. However, expressing nutrient intake as a percentage of requirements also does not cover this important point – the information required is what the nutritional status of the population is. We have added information in the discussion relating to estimated population intake compared to the Estimated Average Requirement (EAR) for calcium (lines 430-433), and Vitamin C (lines 438-440) and guideline recommendations for sugar (lines 383-388).
  6. In general, the authors expected that young children would have a high contribution of beverages to calcium intake because of a high intake of milk and milk based based beverages in particular. The contribution of beverages to calcium was high in each of the adult age groups ranging between 26 and 33% of the total (except for older people 71 years and above where beverages had a lower contribution). We think this reflects intake of milk in tea and coffee for adults, with flavoured milk making a contribution for younger adult groups.
  7. The aim of the survey analysis was to describe beverage intake and its contribution on intake of selected nutrients (lines 399 – 402). We agree that the reasons why people consume beverages in the pattern that they do is important to intervention efforts – however, the authors were not involved in the data collection, and the reasons for particular beverage intake behaviour is likely to be complex (eg lines 365 – 368). The result for tea consumption in remote regions is striking - we have offered a largely speculative explanation in lines 403-405 which relates to Indigenous workers being given tea with their rations (decades in the past). Tea is also relatively easy to store in areas where the power supply may not be reliable. The importance of determining why the behaviour occurs in the way it does is important enough to warrant a specific study for this purpose. We have added some further text to make these issues clearer (lines 405-409). The tea plantation areas in Australia are small and don’t cover the same geographic distribution of the remote and very remote categorisation.
  8. As mentioned above, the authors are reluctant to speculate on the reasons for beverage consumption patterns without direct studies of these. Consumption of alcohol is culturally sensitive and speculation on the reasons for high intake on our part is inappropriate. Reasons for high consumption in particular individuals is unlikely to be simple. Alcoholic drink type (not analysed here) might be relevant to the comparison because it is a weight for weight comparison (not based on alcohol content). We have highlighted that a higher percentage of Indigenous people than non-Indigenous do not drink alcoholic drinks at all.    

Reviewer 2 Report

The manuscript entitled “Beverage intake and associated nutrient contribution for Aboriginal and Torres Strait Islander Australians: Secondary analysis of a national dietary survey 2012-13” by Megan A Rebuli, Gilly A Hendrie, Danielle L Baird, Ray Mahoney and Malcolm D Riley,  sounds interesting for assessing the impact of diet in relation to dysmetabolic risk. Please consider the following points:

Line 18 "Beverage intake contributes to positive and negative nutrient intake for health": better clarify. Line 28: specify the type of beverages, sweetened? alcoholic?

Line 119-122: why are these methods described if the data were not included?

Line 189: unfortunately I cannot see supplementary table 1, please provide it.

The data collected and presented is descriptive only. Please provide at least to introduce a correlation analysis with BMI and pathology. Otherwise, there is very little reader interest in this topic.

Author Response

We thank the reviewers for their comments on our manuscript, and acknowledge the time and thought that has gone into their reviews. We have made a number of amendments to the manuscript that has resulted in its improvement, and we appreciate the contribution from the reviewers.

The authors have once again reviewed the manuscript for English language expression, and have re-numbered the references to include new citations.  

Reviewer 2:

  1. We thank the reviewer for pointing out that this sentence is not clear. While we were trying to highlight that beverage intake makes a substantial contribution to nutrients that can be considered detrimental to health as well as beneficial to health (depending on context), we have determined that the sentence basically restates the summary immediately before it. We have therefore decided it is not required and we have omitted it in our revision.
  2. Our opening introduction sentence is referring to the contribution of all beverage intake for Australians (including alcoholic, sweetened, and all other beverages). We have clarified this by amending to ‘Intake of all beverage types’ at the beginning of this sentence.
  3. The recent national dietary surveys in Australia (including the NATSINPAS) used 2 days of 24 hour recall as the primary dietary intake data source – this is outlined in lines 96-99. However, there are important drawbacks to using both days of measurement in analyses – these are outlined in lines 119-122, justifying why one day of dietary recall is used. We think it is important to provide clarity about the data that was analysed, and request that this text remain.
  4. The supplementary tables are visible to us on the Nutrients website review page – we are unsure about what steps are required to make the tables available also to the reviewer.
  5. We agree with the reviewer that this analysis of dietary survey data is descriptive, but we respectfully disagree that there is little interest in this topic. There is considerable interest among nutritionists and others in the dietary intake of populations. Dietary intake differs across populations for cultural reasons as well as reasons related to food and drink availability. Public health effort to modify dietary intake is acutely focussed on what people currently eat and drink. Our interest in beverage intake is primarily nutrition related - beverage intake is frequently associated with poorer diet quality however we have shown that beverage intake also makes positive contributions to nutrient intake (for Australians, and also for Indigenous Australians). The data is not well suited to assess dietary association with BMI or pathology. The measurement of dietary intake by self-report on a single day does not represent usual (or long-term) dietary intake for individuals, and cross-sectional (survey) data associations makes causality difficult to interpret.  

Round 2

Reviewer 1 Report

Dear authors,

Many thanks for considering the proposed comments and suggestions. In general, the manuscript is improved and in this new version the uniqueness of this survey is highlighted, although it was conducted 9 years ago in 2012-13.

Nevertheless, authors should better justify the results obtained, going behind the speculative explanations. As a result, we ask authors to reanalyze our previous comments, namely:

- Did authors expect the fact that 31-50-year-olds had the greatest percentage contribution of beverages to calcium among adults (32.7%), and only 2-3-year-olds had a higher percentage of calcium coming from beverages (p. 8)?

- Authors should explain why the high consumption of tea in remote living Indigenous people seems to be unique to this population group as it was not observed in metropolitan/regional living Indigenous adults or the survey of all Australians (p. 11). Is there any relation with proximity of tea plantation area?

- Authors should explain why Indigenous people living in remote areas that consumed alcoholic beverages, consumed almost three times as much as Australian adults generally who consumed alcoholic beverages.

This is particularly relevant, as it was reported by authors in the Introduction section that remote zones are faced with the impact of interventions aimed at subsidising and discounting healthier beverages in remote stores such as water and milk while increasing the price of carbonated SSB.

Author Response

We thank the reviewers for their further comments on our manuscript, following our response to their initial comments and our consequent amendments.

Reviewer 1 has asked us to ‘better justify the results obtained, going behind the speculative explanations. As a result, we ask authors to reanalyze our previous comments’ [with comments 6, 7 and 8 from the original review repeated]. The reviewer has not outlined the inadequacy of our previous responses, so we will try to make our response to these comments clearer. In all honesty, we are not able to make any further amendments to the manuscript because we do not understand how we have misunderstood the reviewers previous comments.

Firstly, the purpose of this analysis was to describe the pattern of beverage intake from data collected by a government agency from a national sample of Indigenous Australians for the first and (to date) only time. The survey results are what they are – they cannot be better justified. So we think the reviewer is seeking for us to give an explanation for the results. Dietary survey data is not suitable to determine why behaviour has occurred - only what behaviour has occurred. So an offered explanation for why a beverage pattern is apparent is speculative without further studies. We are reluctant to speculate about reasons for beverage intake behaviour patterns because our proposed explanations may be offensive while serving little purpose – to determine the reasons for difference requires specifically designed and targeted studies. We support the conduct of such studies conditional on the appropriate co-designed and culturally safe involvement and leadership of the communities of Indigenous people. While we have one co-author who is Indigenous, they can not speak for and represent the views of all Indigenous people and communities in Australia.

- Did authors expect the fact that 31-50-year-olds had the greatest percentage contribution of beverages to calcium among adults (32.7%), and only 2-3-year-olds had a higher percentage of calcium coming from beverages (p. 8)?

Our previous response to this comment was lengthy – a shorter response is that the relative contribution of beverages to calcium for 31-50 year olds is consistent to the adjoining age groups and we are aware that beverages make a strong contribution to calcium intake in the youngest age group.

- Authors should explain why the high consumption of tea in remote living Indigenous people seems to be unique to this population group as it was not observed in metropolitan/regional living Indigenous adults or the survey of all Australians (p. 11). Is there any relation with proximity of tea plantation area?

In our previous response to this comment we made the point that tea plantation areas are small and geographically confined in Australia and cannot be an explanation for remote living Indigenous people to have a high tea intake. We pointed out the tentative explanation we had already provided in the manuscript - that tea was given as payment for labour in many remote areas in past times. We offered the further explanation that tea is easily stored in areas where the power supply may be uncertain, and transported where freight costs add to price – the manuscript text was amended to include these points. It is not clear to the authors whether the reviewer rejects these explanations, or has a different expectation of how we respond to the comment made.  Tea, flour, sugar, alcohol and opium were used as “payment” for labour and other purposes in place of cash and have strong historical connection to Australian Government policies of separation, segregation, assimilation and Stolen Generations.  The authors are mindful of the existing and ongoing trauma faced by many Indigenous people in Australia as a result of past Government policies and as such are willing to respectfully make minimum speculation about the relationship between tea consumption.

- Authors should explain why Indigenous people living in remote areas that consumed alcoholic beverages, consumed almost three times as much as Australian adults generally who consumed alcoholic beverages.

The authors are very reluctant to speculate on the almost certainly complex reasons for why Indigenous alcohol drinkers living in remote areas drink more than 3 times the volume of Australian adult alcohol drinkers more generally. Our previous response included: “Consumption of alcohol is culturally sensitive and speculation on the reasons for high intake on our part is inappropriate. Reasons for high consumption in particular individuals is unlikely to be simple. Alcoholic drink type (not analysed here) might be relevant to the comparison because it is a weight for weight comparison (not based on alcohol content). We have highlighted that a higher percentage of Indigenous people than non-Indigenous do not drink alcoholic drinks at all.”

To expand on these points – a high alcohol intake is generally considered to be a negative behaviour with adverse health and social consequences, whereas a high proportion of non-drinkers would be considered to have positive health and social consequences. Excessive alcohol intake is commonly a basis for implied or overt criticism so it would be inappropriate to offer a brief explanation of the survey comparison made by the reviewer for what are very different contexts of living circumstances. To speculate – for Australians generally alcoholic beverages are conveniently available and a large percentage of adults drink alcohol but not much. For many (but not all) Indigenous adults living in remote regions, alcoholic beverages are not conveniently available within communities, and is not kept as a pantry item for cultural as well as community based reasons. Therefore an Indigenous adult alcohol drinker may drink alcohol for the purpose of drinking and therefore drink more because of the inconvenience to obtain supply. The Indigenous people of Australia have been subject to the impact of more than 200 years of colonisation by a numerically dominant society which has had a severe and lasting impact on all aspects of their lives, including social dislocation which might result in heavier drinking. Unemployment among Indigenous adults living in remote locations is generally high allowing more time for consumption of alcoholic drinks. The alcohol content of the beverages consumed might be different (Indigenous adults might consume more beer than wine or spirits impacting on a volume/weight comparison). The reasons for consuming different patterns of alcoholic drinks are very complex and may include considerable variation across the broad categories investigated here. Our approach is to report the facts from the survey data (i.e. the measured beverage intake) and provide comparison across meaningful categories (age group, sex, geographic region) – possible because the survey was conducted.  Alcohol consumption is not the main focus of this paper and presented in isolation and out of context has the potential to re-enforce existing negative stereotypes about Indigenous people are prevalent in the dominant non-Indigenous Australian population.

Reviewer 2 Report

I acknowledge that the authors have made some efforts to improve the quality of the manuscript. The quality data and reader interest remain low. 

Author Response

We thank the reviewers for their further comments on our manuscript, following our response to their initial comments and our consequent amendments.

‘I acknowledge that the authors have made some efforts to improve the quality of the manuscript. The quality data and reader interest remain low.’

The authors are surprised and dismayed at the reviewer comment regarding data quality and reader interest, and must agree to hold a different opinion. The survey data is from a national survey collected by a federal government agency with adequate resource and decades of experience. The dietary data quality, collected from thousands of people has weaknesses common to population survey data but is at least the equivalent of that collected in any country. This national survey was of Australian Indigenous people – for the first time in history. Previous dietary intake surveys of Indigenous people were small, community specific and rarely included Indigenous people living in metropolitan areas. Dietary surveys are relevant to the assessment of dietary health, the development of dietary guidelines and nutrition policy, and the design and monitoring of dietary intervention programs to name a few uses. The authors acknowledge that some readers may have little interest in the dietary intake of Australian Indigenous people (or even perhaps Australians) however we feel that the analytical methodology is strong and includes unique features which might interest readers such as adjusting for day of the week of data collection.  Internationally evidence about dietary intake from First Nations people is very limited (if not non-existent).  We expect there to be a significant interest in this paper internationally and we know anecdotally that academics, researchers, scientist and members of the Indigenous community are keen to see these results.